# Impacts of Captive Domestication and Geographical Divergence on the Gut Microbiome of Endangered Forest Musk Deer

**DOI:** 10.3390/ani15131954

**Published:** 2025-07-02

**Authors:** Huilin Liu, Lu Xiao, Zhiqiang Liu, You Deng, Jinpeng Zhu, Chengzhong Yang, Qing Liu, Di Tian, Xiaojuan Cui, Jianjun Peng

**Affiliations:** 1School of Life and Health Sciences, Hunan University of Science and Technology, Xiangtan 411201, China; lhl1999@mail.hnust.edu.cn (H.L.); 1090039@hnust.edu.cn (L.X.); kdsliu@126.com (Z.L.); dengyouuuaction@163.com (Y.D.); m18973572094@163.com (J.Z.); 22010901023@mail.hnust.edu.cn (Q.L.); tiandi990312@163.com (D.T.); 2Animal Biology Key Laboratory of Chongqing Education Commission of China, Chongqing Key Laboratory of Conservation and Utilization of Freshwater Fishes, College of Life Sciences, Chongqing Normal University, Chongqing 401331, China; drczyang@126.com

**Keywords:** forest musk deer (*Moschus berezovskii* Flerov), GUT microbiota, metagenomic sequencing, geographical divergence, endangered species conservation

## Abstract

Forest musk deer (*Moschus berezovskii* Flerov), globally endangered ruminants, face extinction in the wild, while captive population are further threatened by widespread digestive and immune disorders. This study utilized comparative metagenomic sequencing to analyze gut microbial structures in wild populations (Chongqing and Hunan) and captive individuals. Wild populations exhibited a predominance of Pseudomonadota, featuring enrichment of lignin-degrading genera such as *Novosphingobium* and *Acinetobacter*. Captive population showed a higher relative abundance of Bacillota and Bacteroidota phyla, accompanied by potentially pathogenic overgrowth of *Escherichia* and *Clostridium*. Despite significant richness differences in wild populations due to habitat vegetation, core microbial diversity and carbohydrate metabolic functions exhibited convergence. Captive microbiota showed enriched in translation and sugar metabolism pathways, while wild populations prioritized immune regulation and environmental sensing. These findings provide a theoretical basis for optimizing conservation strategies and for scientifically informed captive management.

## 1. Introduction

Gut microbiota, often described as the host’s “second genome,” profoundly influences ecological adaptation and evolutionary processes through nutrient metabolism regulation [1], immune homeostasis maintenance [2], and growth modulation [3]. Such microbial shifts may serve as biomarkers of population survival status [4]. In ruminants, the efficient decomposition of high-fiber plant material depends on a specialized digestive system synergistically driven by rumen and gut microbiota [5]. Studies demonstrate that multidimensional ecological factors—geographical isolation [6], dietary pattern [7], and habitat vegetation [8]—converge to shape wildlife gut microbiota, with dietary factors frequently dominating [9].

The impact of captive domestication on gut microbiota has garnered significant attention in endangered species conservation. For instance, captive giant pandas (*Ailuropoda melanoleuca* David) exhibit reduced gut microbial diversity and diminished abundance of cellulose-degrading bacteria compared to wild counterpart [10]. Similarly, captive Przewalski’s horses (*Equus ferus przewalskii* Poliakov) display lower gut microbial richness, decreased metabolic gene abundance, and heightened zoonotic disease risks relative to wild populations [11]. These findings suggest that artificial environments—constrained by dietary monotony and limited activity range—may drive irreversible declines in core functional microbial taxa [12]. Geographic variation also plays a pivotal role in shaping wildlife gut microbiota. A systematic analysis of 29 African chimpanzee (*Pan troglodytes* Blumenbach) populations revealed that geographic divergence, mediated by host genetics, vegetation composition, and tool-use-associated dietary differences, drives significant regional microbial community differentiation [13]. Comparative studies of captive Asiatic black bear (*Ursus thibetanus* G. Cuvier) across Sichuan, Yunnan, and Heilongjiang provinces further demonstrated geographically driven divergence in gut microbial diversity, dominant taxa, and community structure, implicating latitude-associated climatic factors in host–microbe coevolution [14].

The forest musk deer (*Moschus berezovskii* Flerov), a small ungulate endemic to East Asia, is classified as a National Grade I Protected Species in China and listed as endangered under CITES Appendix II and the IUCN Red List [15,16]. Its unique musk secretion capability and adaptation to alpine shrublands have shaped gut microbiota with specialized secondary metabolite conversion capacities, endowing ecological, scientific, and economic significance [17,18]. However, rampant poaching and habitat degradation have driven severe population on declines, critically threatening species survival [19]. Current research prioritizes disease prevention, musk secretion mechanisms, and genetic diversity [20,21,22], while critical gaps persist regarding in understanding geographic variability of wild populations and microbial functional adaptation during captivity. This study applies metagenomic sequencing to compare gut microbiota between wild (Chongqing and Hunan) and captive populations, providing microbiome-driven insights to enhance habitat management and optimize captive breeding protocols for evidence-based conservation strategies.

## 2. Materials and Methods

### 2.1. Sample Collection

This study strictly followed to non-invasive sampling protocols, with all fecal samples collected from naturally deposited feces of forest musk deer without animal capture, sedation, or direct contact. Given the critically endangered status of forest musk deer (*Moschus berezovskii* Flerov) and their exceptionally low detectability in wild habitats, obtainable fecal samples under natural conditions remain extremely scarce. Our sampling was consequently restricted to *n* = 8 specimens per group from both wild and captive breeding populations. While this limited sample size constrains population-level inferences, it enables high-resolution profiling of individual gut microbiome adaptations through metagenomic analysis under managed conditions. Wild samples were obtained from Jinfo Mountain National Nature Reserve, Chongqing (Wild Group W1: 107°03′–107°26′ E, 28°46′–29°30′ N, *n* = 8) and Huping Mountain National Nature Reserve, Hunan (Wild Group W2: 110°29′–110°59′ E, 29°58′–30°08′ N, *n* = 8). Wild sampling protocols required a (1) selection of fresh, large fecal pellets to ensure collection from adult individuals and (2) spatial separation of samples by >500 m to minimize home-range overlap (2.8–7 hm^2^ per individual), guaranteeing distinct host origins [23]. Captive samples originated from Yongshun Mingfa Musk Deer Farm, Xiangxi, Hunan (Captive Group C: 110°58′ E, 29°01′ N, *n* = 8), where semi-open enclosures with individual pens were maintained. Fresh captive faces were collected during morning cleaning (detailed sample information can be found in Appendix A). All samples were immediately transferred to sterile cryovials containing 80% ethanol (pre-cooled to −20 °C) using aseptic gloves within 30 min post-collection, transported at 4 °C, and stored at −80 °C until analysis.

### 2.2. Morphological and Molecular Biological Identification of Samples

#### 2.2.1. Fecal Morphological Analysis

In field surveys, forest musk deer (*Moschus berezovskii* Flerov) feces are identified as plump, olive-shaped pellets, typically dark brown to blackish-brown, with a smooth, glossy surface; they measure approximately 3–5 mm in diameter and 7–9 mm in length, and fresh male specimens may exhibit a faint musk odor. For comparison, tufted deer (*Elaphodus Cephalophus* Milne-Edwards) feces are larger with a distinct inward indentation but less plump, whereas Reeves’s muntjac (*Muntiacus reevesi* Ogilby) feces appear more rounded, featuring a moldy bluish-green surface interspersed with yellowish streaks, and tend to clump due to roughness. Serow (*Capricornis sumatraensis* Bechstein) feces are oval-shaped, dark-colored, and moistly glossy; gorals (Naemorhedus goral) produce irregular, loosely formed pellets with black and yellow mottling; red-and-white Giant (*Petaurista alborufus* Milne-Edwards) excrete distinct, cube-shaped, yellowish-brown pellets; goats (*Capra aegagrus hircus* Linnaeus) deposit oval, black feces akin to soybean size; and cattle (*Bos taurus* Linnaeus) or sheep (*Ovis aries* Linnaeus) feces are generally larger and more variable in color.

#### 2.2.2. Molecular Biological Identification: DNA Extraction and PCR Amplification

Total DNA was extracted from fecal samples using the QIAamp Fast DNA Stool Mini Kit (Qiagen, Hilden, Germany). Species-specific primers were designed with Primer3 online software (V2.4.0) based on the complete mitochondrial genome of *Moschus berezovskii* Flerov (GenBank accession: PP025305). The primer sequences were as follows:
MBshortF4: 5′-TAGGTTAAATAGACCAAGAGCCTTCA-3′;MBshortR4: 5′-AGTTCGGCACGGATTAGCAG-3′.

PCR amplification was performed in a 20 μL reaction mixture containing 10 μL of 2× Taq PCR Master Mix (Tiangen Biotech, Beijing, China), 1 μL each of forward and reverse primers (10 μM), 1 μL template DNA, and nuclease-free water to adjust the final volume to 20 μL. The thermal cycling protocol comprised initial denaturation at 95 °C for 3 min; 35 cycles of denaturation at 95 °C for 30 s, annealing at 58 °C for 30 s, and extension at 72 °C for 30 s; followed by a final extension at 72 °C for 5 min with terminal holding at 4 °C.

PCR products (5 μL aliquots) were analyzed by 1% agarose gel electrophoresis. Amplicons matching the expected size were purified and subjected to bidirectional Sanger sequencing at Sangon Biotech (Shanghai, China). Raw sequence data were assembled and proofread using SeqMan software (V17.2), then deposited into the NCBI database. Species identification was confirmed through homology alignment via the BLAST online tool (https://blast.ncbi.nlm.nih.gov/Blast.cgi, accessed on 25 November 2024), with sequence similarity analysis determining taxonomic origin.

### 2.3. DNA Extraction and Quality Control

Total genomic DNA was extracted from fecal samples using the TIANamp Stool DNA Kit (TIANGEN, Beijing, China) following manufacturer protocols. DNA integrity was assessed via 1% agarose gel electrophoresis to verify absence of degradation and contaminants. Purity was quantified using a NanoDrop 2000 spectrophotometer (IMPLEN, Westlake Village, CA, USA) for absorbance ratios, while concentrations were determined via Qubit^®^ 2.0 Fluorometer with the Qubit^®^ dsDNA HS Assay Kit (Thermo Fisher Scientific, Waltham, MA, USA).

### 2.4. Library Preparation and Sequencing

Genomic DNA (1 μg per sample) was fragmented into ~350 bp inserts using a Covaris ultrasonicator (Covaris, Woburn, MA, USA). Libraries were constructed with the NEB Next^®^ Ultra™ DNA Library Prep Kit (NEB, Ipswich, MA, USA) through end repair, adapter ligation, size selection, and PCR amplification. Library quality was verified by Agilent 2100 Bioanalyzer (Agilent, Santa Clara, CA, USA) for insert size distribution and quantified via qPCR (effective concentration > 3 nM). Paired-end sequencing (2 × 150 bp) was performed on the Illumina NovaSeq 6000 platform at Novogene Bioinformatics Technology Co., Ltd. (Tianjin, China). Raw sequencing data were deposited in the NCBI Sequence Read Archive under accession PRJNA1242249.

### 2.5. Bioinformatic Analysis

#### 2.5.1. Data Preprocessing

Raw reads were filtered using fastp (https://github.com/OpenGene/fastp, accessed on 29 June 2025) with three criteria: (a) removal of read pairs with adapter contamination; (b) exclusion of sequences with >50% low-quality bases (Q ≤ 5); and (c) elimination of reads containing ≥10% ambiguous N bases. Potential host DNA contamination was mitigated by aligning reads to the musk deer reference genome using Bowtie2 (V2.5.4) (http://bowtie-bio.sourceforge.net/bowtie2/index.shtml, accessed on 29 June 2025; parameters: –end-to-end, –sensitive, –I 200, –X 400) [24,25]. Clean data were validated via FastQC for downstream analyses.

#### 2.5.2. Metagenomic Assembly and Gene Catalog Construction

Clean data were assembled using MEGAHIT (--presets meta-large) to optimize microbial community reconstruction. Scaffolds were fragmented at N-base junctions into non-overlapping scaftigs (≥500 bp) [26]. Open reading frames (ORFs) were predicted via MetaGeneMark (http://topaz.gatech.edu/GeneMark/, accessed on 29 June 2025), with sequences < 100 nt discarded [27]. Redundant genes were clustered at 95% similarity using CD-HIT (-c 0.95, -G 0, -aS 0.9, -g 1, -d 0) to generate a non-redundant gene catalog [28]. Gene abundance was calculated by mapping clean reads to the catalog using Bowtie2, retaining genes with ≥2 reads [29]. Core–pan gene analysis, inter-sample correlations, and Venn diagram generation were conducted based on gene abundance profiles.

#### 2.5.3. Taxonomic Profiling and Functional

Taxonomic annotation was performed using DIAMOND software (v2.1.12) (https://github.com/bbuchfink/diamond/, accessed on 29 June 2025; parameters: blastp, −e 1 × 10^−5^) to align non-redundant gene sets (unigenes) against the Micro_NR database, which integrates bacterial, fungal, archaeal, and viral sequences from the NCBI NR database (https://www.ncbi.nlm.nih.gov/, accessed on 29 June 2025) to comprehensively capture environmental microbial diversity [30,31]. MEGAN’s LCA algorithm (https://en.wikipedia.org/wiki/Lowest_common_ancestor, accessed on 29 June 2025) assigned taxonomic ranks from phylum to species, with abundances derived from annotated gene counts [31]. α-diversity indices (Shannon, Ace, Chao1, Simpson) were compared across groups via Kruskal–Wallis tests. β-diversity was assessed using unweighted/weighted UniFrac distances visualized via PCoA and heatmap clustering. Differentially abundant taxa and pathways were identified via MetaGenomeSeq (R, V3.4.1) and LEfSe (V1.0) (LDA ≥ 2) [32]. Predictive biomarkers were selected using RandomForest (R packages pROC and randomForest, v2.15.3) [33]. Functional annotation employed DIAMOND against KEGG, with pathway abundances normalized to gene counts [34]. Metabolic pathway variations were evaluated using LEfSe and MetaGenomeSeq.

## 3. Results

### 3.1. Gut Microbial Community Composition

Metagenomic sequencing analysis of 24 fecal samples from forest musk deer (*Moschus berezovskii* Flerov) generated 201.03 Gb of raw data using the Illumina platform. After quality control, 198.49 Gb of high-quality sequences were retained (average Q20: 98.65%, Q30: 96.08%, GC content: 55.29%, effective sequence rate: 98.73%). Sequencing depth reached saturation (rarefaction curve shown in Appendix A), meeting the requirements for subsequent analyses (detailed QC data are provided in Appendix A). At the phylum level (Figure 1a), the dominant phyla in the gut microbiota of the Chongqing wild population (W1) were *Pseudomonadota* (50.78 ± 0.22%), *Bacillota* (15.69 ± 0.22%), *Actinomycetota* (10.29 ± 0.05%), *Bacteroidota* (5.93 ± 0.04%), and *Myxococcota* (1.07 ± 0.02%). In Hunan wild population (W2), the predominant phyla were *Pseudomonadota* (51.60 ± 0.22%), *Bacillota* (6.12 ± 0.13%), *Bacteroidota* (5.87 ± 0.03%), *Myxococcota* (5.36 ± 0.08%), and *Acidobacteriota* (2.86 ± 0.02%). For captive forest musk deer, the dominant phyla included *Bacillota* (32.12 ± 0.24%), *Pseudomonadota* (29.26 ± 0.29%), *Actinomycetota* (6.91 ± 0.07%), *Bacteroidota* (4.09 ± 0.03%), and *Candidatus saccharibacteria* (1.03 ± 0.01%).

At the genus level (Figure 1b), the gut microbiota of W1 was dominated by Acinetobacter (11.37 ± 0.19%), *Novosphingobium* (3.88 ± 0.06%), *Comamonas* (1.71 ± 0.03%), Stenotrophomonas (1.51 ± 0.04%), and *Sphingopyxis* (1.37 ± 0.02%). In the W2 group, the predominant genera were *Novosphingobium* (3.62 ± 0.04%), *Mesorhizobium* (3.58 ± 0.06%), Acinetobacter (2.75 ± 0.06%), *Sphingomonas* (2.38 ± 0.03%), and *Luteimonas* (1.41 ± 0.02%). Captive forest musk deer exhibited dominance by *Acinetobacter* (8.4 ± 0.12%), *Escherichia* (3.62 ± 0.05%), *Clostridium* (2.70 ± 0.02%), *Enterococcus* (1.87 ± 0.03%), and *Stenotrophomonas* (1.06 ± 0.03%). Venn diagram analysis (Figure 1c) revealed that wild forest musk deer (W1 and W2 groups) harbored more unique microbial species than the captive group. The W2 group had a greater number of unique species compared to the W1 group, and the number of species shared between W1 and W2 groups far exceeded those shared between wild and captive groups.

### 3.2. Comparative Analysis of Gut Microbial Diversity and Differences

Systematic analysis of α-diversity (ACE, Chao1, Shannon, and Simpson indices) and β-diversity metrics was conducted to evaluate gut microbiota diversity and community evenness across the three groups (alpha diversity statistics in Appendix A). The Good’s coverage index for all samples exceeded 0.99 (Appendix A), confirming sequencing data representativeness. Results (Figure 2a) demonstrated that wild groups (W1, W2) exhibited significantly higher ACE and Chao1 indices than the captive group (C), indicating superior species richness in wild populations. While ACE indices differed significantly between W1 and W2 groups, Shannon and Simpson indices showed no significant variation, suggesting divergent microbial community abundances but convergent species diversity between the two wild groups.

Principal Coordinate Analysis (PCoA) (Figure 2b) revealed significant β-diversity differences among the three groups. PCoA1 and PCoA2 accounted for 36.54% and 24.48% of the sample variation, respectively, cumulatively explaining 61.02% of the total variation. A clear separation was observed between the captive group (C) and the Hunan wild group (W2), indicating significant gut microbiota composition differences between captive individuals and their Hunan wild counterparts. In contrast, the Chongqing wild group (W1) exhibited overlap with both W2 and C groups, suggesting structural similarities in gut microbiota among W1, W2, and C. Additionally, partial spatial segregation between Chongqing (W1) and Hunan (W2) wild groups indicated distinct gut microbiota compositions.

Cluster analysis of the top 12 relatively abundant taxa with significant intergroup differences was visualized via heatmap (Figure 2c). Differentially abundant species were identified using MetaGenomeSeq and Kruskal–Wallis rank-sum tests. Boxplots (Figure 3) illustrated the distribution of the top 12 genera across groups. Wild groups (W1, W2) showed significantly higher abundances of *Advenella*, *Brevundimonas*, *Chitinophaga*, *Oerskovia*, *Hydrogenophaga*, *Microbacterium*, *Mesorhizobium*, *Novosphingobium*, and *Sphingopyxis* compared to the captive group (C) (*p* < 0.05). In contrast, *Clostridium*, *Enterococcus*, and *Glutamicibacter* were enriched in the captive group (*p* < 0.05). *Microbacterium* abundance differed significantly between W1 and W2 groups. (*p* < 0.05).

### 3.3. Functional Composition of Gut Microbiota

KEGG annotation revealed gut microbiota functional genes were primarily distributed among six core metabolic hierarchies. Metabolism (57.3%) was the dominant functional module, with carbohydrate metabolism and amino acid metabolism serving as central components. Subsidiary categories included Environmental Information Processing (11.5%), Genetic Information Processing (10.9%), Cellular Processes (9.1%), Human Diseases (7.5%), and Organismal Systems (3.7%) (Figure 4a). LEfSe analysis (LDA ≥ 2, Figure 4b) identified group-specific functional pathways: the captive group (C) was enriched in translation, glycan biosynthesis and metabolism, and nucleotide metabolism; Group W1 showed dominance in sensory systems, eukaryotic cellular communities, and immune system pathways; Group W2 exhibited associations with cell motility, energy metabolism, metabolism of other amino acids, cardiovascular diseases, endocrine/metabolic diseases, and aging. Clustered heatmaps (Figure 4c) further illustrated abundance patterns of differentially abundant metabolic pathways across samples.

## 4. Discussion

This study compared gut microbiota composition between wild forest musk deer (*Moschus berezovskii* Flerov) from Chongqing (W1) and Hunan (W2) with that of captive individuals (C) using metagenomic data, revealing that geographical factors and living environment significantly shape intestinal microbial structure. At the phylum level, Pseudomonadota, Bacillota, and Bacteroidota were dominant across all groups. Wild groups (W1, W2) exhibited higher Pseudomonadota abundance than the captive group (C), while Bacillota predominated in captivity, aligning with previous findings on wild–captive microbiota divergence in forest musk deer [35]. Pseudomonadota in ruminant gut microbiota modulates host immune function through symbiotic metabolism, particularly via short-chain fatty acids (SCFAs) produced during high-fiber degradation. These SCFAs enhance intestinal barrier integrity, suppress inflammation, and improve energy extraction from low-nutrient vegetation [36,37]. The food sources for captive populations primarily consist of artificially supplied feed. Based on our research, their diet exhibits high monotony, typically limited to hand-picked leaves (e.g., mulberry, plum, and paper mulberry) supplemented with standardized concentrate feed. Additionally, probiotics and other biological supplements are routinely incorporated during artificial feeding. In contrast, wild populations derive nutrition from significantly diverse sources, including tree leaves, roots, berries, medicinal plants, and other naturally available resources. This substantial divergence in dietary composition between captive and wild groups indicates that differences in their gut microbiota are predominantly attributable to variations in nutritional intake. Proteobacteria’s prevalence on soil and plant surfaces may further amplify colonization opportunities in wild forest musk deer, as its abundance correlates with natural vegetation consumption [38]. Notably, Bacillota abundance further diverged markedly between two wild populations (W1: 15.69% vs. W2: 6.12%), reflecting its role in cellulose metabolism. This phylum employs enzymatic systems to degrade dietary fibers into SCFAs (e.g., acetate, propionate, butyrate), enhancing nutrient absorption efficiency [39]. The elevated Bacillota levels in captive groups—significantly exceeding wild populations—are linked to metabolic disorders, likely driven by captive diets and conditions [40]. Bacteroidetes contributes to gut ecosystem stability and immunity through carbohydrate-active enzymes that degrade substrates into SCFAs [39,41], though reduced Bacillota-to-Bacteroidetes ratios are implicated in diarrheal pathogenesis [42].

At the genus level, *Novosphingobium* and *Acinetobacter* dominated wild groups (W1, W2), both exhibiting lignin-degrading capabilities [43,44]. These genera facilitate the conversion of wild vegetation into bioavailable carbon sources [45,46], aiding plant metabolites and environmental adaptation. Regional divergence was observed in *Mesorhizobium* abundance—a genus traditionally associated with nitrogen fixation in legume root symbiosis [47]. Its prominence in the W2 group suggests a legume-rich diet shaped by local vegetation and soil microbiota, paralleling geographical drivers of gut microbiota variation in baboons [48]. In contrast, *Escherichia*, *Clostridium*, and *Enterococcus* dominated captive individuals. While *Escherichia* maintains gut microecological balance, its overgrowth under dysbiosis may trigger gastrointestinal diseases [49], its pathogenic role in captive diarrhea [50]. *Clostridium*, exhibits probiotic potential by mitigating inflammation and supporting immune homeostasis [51,52], though its growth is suppressed during intestinal inflammation [53,54]. Additionally, *Clostridium* contributes crucially to cellulose and hemicellulose digestion [55]. *Enterococcus* demonstrates antimicrobial activity and probiotic potential, reducing antibiotic-associated diarrhea and enhancing immunity in captive diets [56,57]. These intergroup differences likely reflect dietary disparities.

α- and β-diversity analyses confirmed significantly higher gut microbiota richness in wild groups than in captive, consistent with prior studies [10,23,58]. Wild groups consume diverse natural vegetation (herbs, woody plants, vines), whereas captive diets are homogenized, reducing microbial diversity [59]. Diet-driven microbiota restructuring is well-documented across species [7,8,60]. Between W1 and W2, microbiota abundance differed markedly, but α-diversity remained comparable. The Chongqing population (W1) inhabits the Dalou Mountains (1900–2200 m elevation; subtropical humid monsoon climate; 1400 mm annual rainfall), favoring evergreen broad-leaved forests and rocky cliffs above 1600 m. In contrast, the Hunan population (W2) resides in the Wuling Mountains (800–1800 m; >1800 mm rainfall), occupying shrub-rich evergreen broad-leaved forests above 1200 m. Dietary analysis (Appendix A) revealed site-specific preferences: Chongqing individuals consumed medicinal plants/berries, while Hunan individuals foraged on foliage/berries. These ecological contrasts in altitude, vegetation, and nutrition drove habitat-specific microbial abundance patterns. Conserved α-diversity between populations suggests evolutionary maintenance of core microbiome functionality critical for metabolic pathways and intestinal homeostasis [61,62], potentially reinforced by phylogenetic constraints on metabolic network regulation [63]. These findings collectively highlight how geographical segregation differentially modulates microbiome abundance while maintaining diversity through host–microbe coevolutionary mechanisms. KEGG notation highlighted gut microbiota genes’ involvement in six metabolic hierarchies, with “Metabolism” (57.3%) dominating—particularly carbohydrate/amino acid metabolism, central to deer physiology [64]. Enrichment of pathways like translation, glycan biosynthesis, and nucleotide metabolism in captive individuals likely reflects high-carbohydrate, protein-rich artificial diets [65]. In contrast, wild groups’ functional distinctions in immunity, sensory systems, and cardiovascular pathways underscore geography-driven microbial divergence [66,67,68].

## 5. Conclusions

Overall, dietary variations and habitat environments partially influence the intestinal microbiota of *Moschus berezovskii* Flerov. Geographic differences between Chongqing and Hunan populations lead to distinct primary food resources due to divergent habitats, resulting in microbiota variations between groups—although our study found these differences statistically insignificant. This likely stems from core microbial communities conserved within the species, which maintain essential intestinal functions regardless of local ecological disparities. Potential mechanisms include convergent core microbiota and conserved genetic regulation of baseline metabolic pathways, meriting future investigation. For captive populations, long-term reliance on uniform artificial diets under confined conditions compromises physiological immunity, whereas wild individuals consume diverse natural vegetation, fostering stable microbiota through evolutionary adaptation. The prevalent gastrointestinal disorders observed in captive forest musk deer may be associated with dietary monotony and/or captive environments, though definitive causality requires additional research.

Nevertheless, this study has limitations, including a sample size that may limit result generalizability and an incomplete assessment of long-term impacts from dietary and geographical factors on the gut microbiome. Future research should expand sample sizes and integrate multidimensional ecological factors for a more comprehensive understanding of functional dynamics in the forest musk deer gut microbiota. Regarding conservation and management, we recommend the following:(1)Optimizing captive environments by simulating natural habitat vegetation and dietary diversity to mitigate declines in gut microbial diversity;(2)Enhancing monitoring of pathogenic bacteria in captive individuals to develop scientific health management protocols;(3)Promoting data sharing and collaborative research across geographic regions to better understand and protect the gut microbial communities of this endangered species.

## Figures and Tables

**Figure 1 animals-15-01954-f001:**
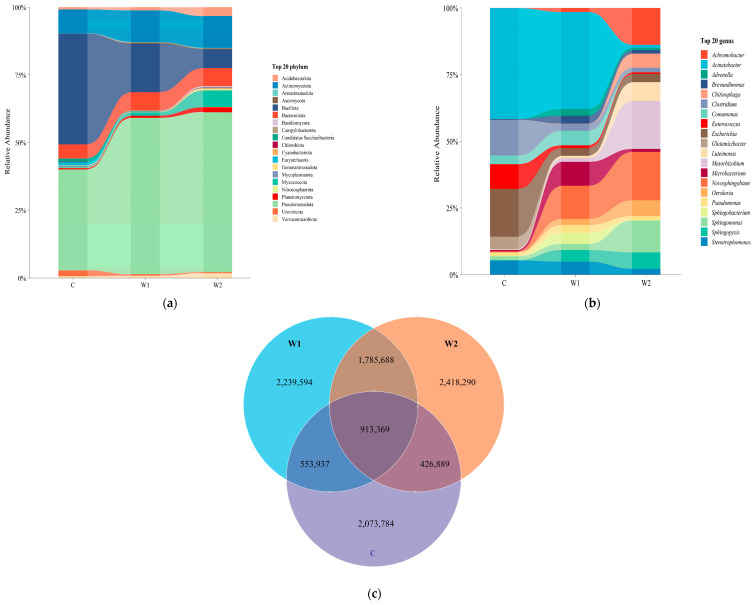
Compositional characteristics of fecal microbiota in forest musk deer (*Moschus berezovskii* Flerov). (**a**) Phylum-level abundance distribution: Stacked bar charts display the relative abundances of the top 20 bacterial phyla. (**b**) Genus-level abundance distribution: Stacked bar charts illustrate the relative abundances of the top 20 bacterial genera. (**c**) Intergroup species distribution: Venn diagram quantifies shared/unique species among the three groups, where overlapping regions represent shared taxa and non-overlapping areas indicate group-specific taxa (W1: Chongqing wild group; W2: Hunan wild group; C: captive group).

**Figure 2 animals-15-01954-f002:**
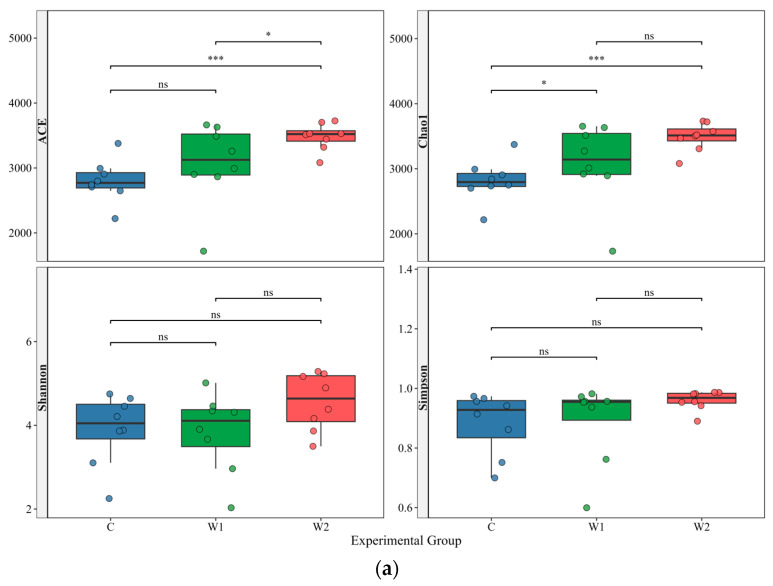
Analysis of diversity and structural characteristics of gut microbiota in different groups of forest musk deer (*Moschus berezovskii* Flerov). (**a**) Comparison of α-diversity indices of gut microbiota among the three groups. Significance levels are denoted as “*” *p* < 0.05, “***” *p* < 0.001 (Kruskal–Wallis rank-sum test), and “ns” indicates no significant difference. (**b**) Principal Coordinate Analysis (PCoA) of β-diversity in fecal microbiota. (**c**) Clustered heatmap of the top 12 most significantly differentially abundant genera at the genus level (W1: Chongqing wild group; W2: Hunan wild group; C: captive group).

**Figure 3 animals-15-01954-f003:**
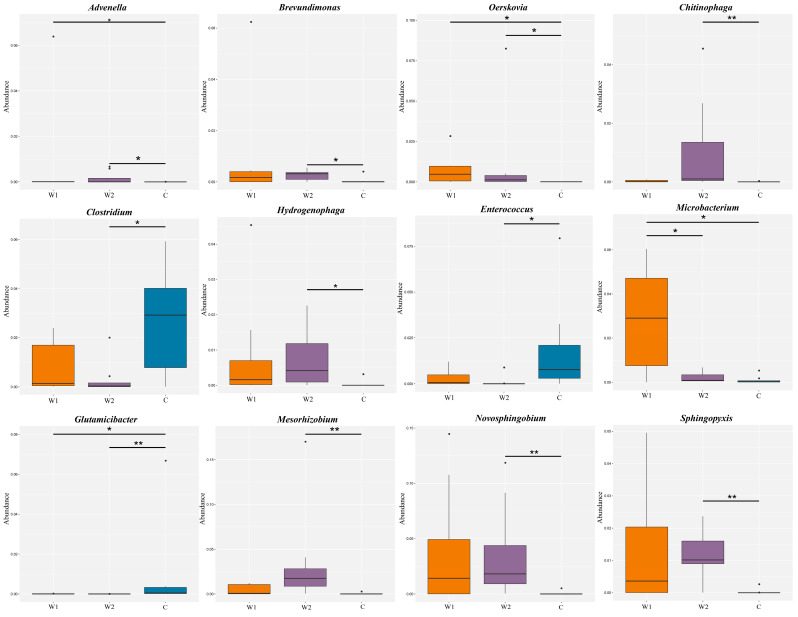
MetaGenomeSeq analysis of gut microbiota at the genus level in three groups of forest musk deer (*Moschus berezovskii* Flerov). The horizontal axis represents sample groups, while the vertical axis indicates the relative abundance of corresponding microbial taxa. Horizontal bars denote significant differences between two groups; absence of a bar implies no significant difference for that taxon. Significance levels are marked as “*” (*p* < 0.05) and “**” (*p* < 0.01) based on the Kruskal–Wallis rank-sum test (W1: Chongqing wild group; W2: Hunan wild group; C: captive group).

**Figure 4 animals-15-01954-f004:**
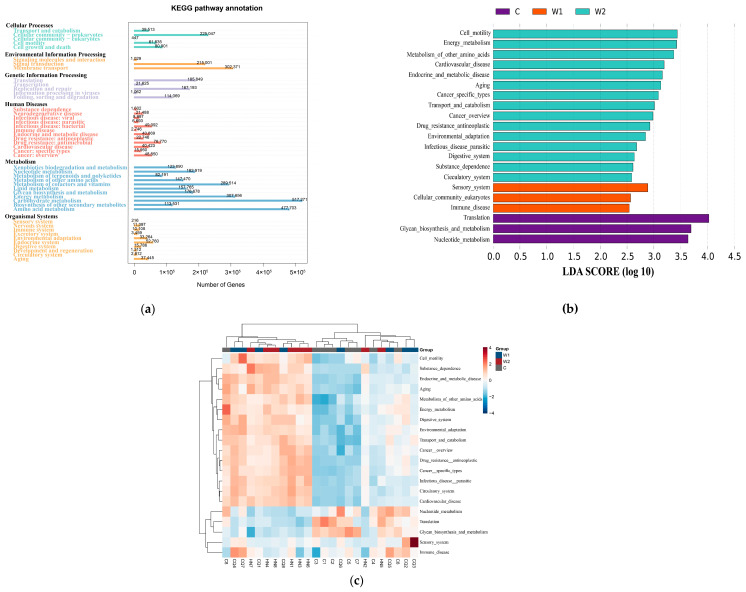
KEGG annotation and LEfSe analysis. (**a**) Statistical distribution of annotated gene counts in the KEGG database. (**a**) The bar plot illustrates the number of annotated genes at the Level 1 hierarchy of the KEGG database based on unigene annotation results. (**b**) LDA score distribution of differential functions (Level 2, LDA = 2). The histogram highlights pathways with significant functional divergence (linear discriminant analysis [LDA] score threshold ≥ 2), emphasizing group-specific metabolic specialization. (**c**) Clustered heatmap of differentially enriched functional abundances in KEGG pathways (Level 2). The heatmap visualizes the relative abundance patterns of functionally distinct pathways across experimental groups (W1: Chongqing wild group; W2: Hunan wild group; C: captive group).

## Data Availability

The original contributions presented in this study are included in this article. Further inquiries can be directed to the corresponding authors.

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
