# Peer review of "Impacts of Captive Domestication and Geographical Divergence on the Gut Microbiome of Endangered Forest Musk Deer"

_animals, 2025, doi:10.3390/ani15131954_

Round 1
Reviewer 1 Report
Comments and Suggestions for Authors
- Why choose forest musk deer populations from Chongqing and Hunan instead of more populations with different geographical distributions? What are the special characteristics of the populations in these two regions?
- In section 2.1, 8 samples of forest musk deer feces were collected from each of the 3 groups, which is a relatively small sample size. Furthermore, how do you determine if the collected feces are from forest musk deer? How do you determine if the collected fecal samples belong to different individuals?
- In section 2.1, it is necessary to describe the sampling time and explain the habitat environment of the three groups of forest musk deer, such as altitude, dominant vegetation, and main feeding plants. For captive populations, is the food source of forest musk deer free feeding or artificial feeding?
- Why is the “Number of samples” in FIGURE S1 23 instead of 24?
- The resolution of FIGURE 1 is too low, especially for the legend, making it difficult to read. All the figures have the problem of low resolution, which makes it difficult to read and recognize. It is recommended to adjust them.
- From FIGURE 2b, it can be seen that only the capstive group (C) and Hunan (W2) wild group show significant separation, and further explanation is needed for lines 203-207.
- In the discussion, it is necessary to add an analysis of the habitat environment of two populations in Chongqing and Hunan, as well as their impact on gut microbiota. For example, whether there are differences in edible plants (dietary habits) between two different geographical regions, whether there are significant differences in climate. Are the differences in gut microbiota caused by these factors?
- Is the food source for captive populations free to eat or artificially fed? Is the difference between captive and wild populations caused by artificial feeding of food?
- Compared with the captive group, the wild group showed a higher abundance of Pseudomonas aeruginosa (lines 257-258), and further analysis is needed to determine the reasons for this result.
- Bacillota is more prevalent in captive individuals (line 259), and there is a significant difference in Bacillota abundance between the two wild populations (lines 264-265), which requires further analysis of the reasons.
- The Conclusions section needs further modification and streamlining. Lines 337-346 are limitations of the paper.
- Carefully examine the entire text, as there are multiple instances of non-standard usage in the text. For example, “P” should be italicized. In section 3.3, the article should annotate Figure 4a. In line 237, 'Figure 4B-C' should be 'Figure 4b and c'.
- Carefully review the references. For example, the Latin names of species in multiple references are not italicized. The 15th reference appears in Chinese. Multiple references do not meet the format requirements of the journal, such as whether to add a dot after the journal abbreviation. Multiple references have non-standard punctuation, such as missing spaces in lines 413 and 523.
Reviewer 2 Report
Comments and Suggestions for Authors
Dear Authors,
Your manuscript titled "Impacts of Captive Domestication and Geographical Divergence on the Gut Microbiome of Endargered Forest Musk Deer" is addressing an interesting topic of research. It might be, therefore, suitable for publication in Animals after major revision. Please, see below a list of comments/suggestions to be applied by you before accepting it.
Yours sincerely,
Reviewer.
L14, L25, L41, L42, L73, L83, L90, L158, L185, L218, L224, L253-L254 and L316: Use scientific nomenclature to cite "Moschus berezovskii Flerov".
L15, L28-L29, L84, L92-L93, L164, L166, L172, L180, L181-L183, L189-L190, L199, L206, L216, L221, L228, L238, L240, L250, L254, L278, L283, L301, L316, L326 and L340: Use scientific nomenclature to refer to Chonqing and Hunan. Specify if both populations are recognized varities from Moschus berezovskii Flerov.
L18 and L20: Use scientific nomenclature to cite those species appropriately.
L30-L32: Use scientific nomenclature to cite those species appropriately.
L37: Replace "pathways, Captive" by "pathways. Captive".
L58-L59: Use scientific nomenclature to cite this specie appropriately.
L61: Use scientific nomenclature to cite this specie appropriately.
L66: Use scientific nomenclature to cite this specie appropriately.
L69-L70: Use scientific nomenclature to cite this specie appropriately.
L70: Write "Sichuan, Yunnan, and Heilongjiang" without italics.
L89-L94 and L97-L100: Provide a full description of the animals under study. Tell how many are in each place, which is the average age, BW, BCS, DMI, etc. nay available information to characterize them. Add also a description of the typical food ingredients used in their diets and its proportion.
L92, L94 and L99: Specify the main criteria used for setting your experimental design. Give full details why are you sampling only 8 samples per treatment. Which program do you use to fix that? Tell us how treatments were assigned and randomization of samples was made.
L94-L97: Give full details about the sampling protocol. Specify how are you sure that samples are not belonging to the same individue. Please, add a table in which it's summarized the location, date of collection, year, hour, and coordenates, etc. for each sample collected.
L172-L179: Use scientific nomenclature to cite all these species appropriately.
Figure 1: Increase its size to gain visibility. In the current state nothing can be clearly distinguished. It's necessary that readers be able to see its content. Please correct it appropriately.
L212-L215: Use scientific nomenclature to cite all these species appropriately.
Figure 2: Increase its size to gain visibility. In the current state nothing can be clearly distinguished. It's necessary that readers be able to see its content. Please correct it appropriately.
Figure 3: Increase its size to gain visibility. In the current state nothing can be clearly distinguished. It's necessary that readers be able to see its content. Please correct it appropriately.
Figure 4: Increase its size to gain visibility. In the current state nothing can be clearly distinguished. It's necessary that readers be able to see its content. Please correct it appropriately.
L277-L293: Use scientific nomenclature to cite all these species appropriately.
L252-l313: Verify if more papers are also available from gut microbiome of musk deer.
L315-L346: Try to be concise. Don't add Results to your Conclusions. Provide new ideas and give advices and/or recommendations for future practical work to be implemented in short-term.
L374-L544: Check your References and verify that all of them are cited according to Animals' instructions for authors. Review each one separately to see in the equirements are followed.
Round 2
Reviewer 1 Report
Comments and Suggestions for Authors
The authors have carefully revised and provided point-to-point responses. However, there are still some minor issues that need further modification. After further revisions, the manuscript can be considered for acceptance.
- In Response 2, the authors mentioned that they employed a dual verification approach using both molecular biology (genetic analysis) and morphology to identify the species origin of the feces, and it is recommended to include this sentence in Section 2. In addition, the molecular biology (genetic analysis) methods mentioned should be described in detail.
- In line 91, there are some minor errors. Should 'anima' be 'animal'? The 'r' in 'deer' should not be italicized.
- Response 8 is beneficial for readers' understanding of the manuscript, and it is recommended to add it to the manuscript.
Reviewer 2 Report
Comments and Suggestions for Authors
Dear Authors,
Your manuscript might be now suitable for publication in Animals after minor revisions. Please, see below a list of comments/suggestions to be addressed by you before accepting it.
Yours sincerely,
Reviewer.
- Add a brief explanation to the manuscript of why such a small number of samples was used, with the justification provided in the revision, so that it is available to readers of the article. Try to explain them why 8 samples from the individuals were only investigated.
- Add a brief explanation to the article of the sampling protocol provided in the revision.
- Use scientific nomenclature to cite Primary Food Source in Table S1. For example, replace 'Eryngium foetidum' by 'Eryngium foetidum L.' and 'Taraxacum mongolicum Hand.-Mazz.' by providing the person who first identified each species.
- Check the website https://www.ncbi.nlm.nih.gov/taxonomy for correct taxonomy citation of species or any other available for scientific use.
- Print Figures and check that images and letters have enough visibility for potential readers.
